# Institutional capacity assessment in the lens of implementation research: Capacity of the local institutions in delivering WASH services at Cox's Bazar district, Bangladesh

Mahbubur Rahman[1]*, Mahbub-Ul Alam[1], Sharmin Khan Luies[1], Sharika Ferdous[1], Zahidul Mamun[2], Musarrat Jabeen Rahman[3], Debashish Biswas[1], Tazrina Ananya[4], Asadullah[1], Abul Kamal[1], Ritthick Chowdhury[5], Eheteshamul Russel Khan[5], Dara Johnston[6], Martin Worth[7], Umme Farwa Daisy[1], Tanvir Ahmed[4,8]

1 Environmental Health and WASH, Health System and Population Studies Division, icddr,b, Dhaka, Bangladesh, 2 WASH Section, UNICEF, Cox's Bazar, Bangladesh, 3 Johns Hopkins Bloomberg School of Public Health, Baltimore, Maryland, United States of America, 4 International Training Network (ITN), Bangladesh University of Engineering and Technology (BUET), Dhaka, Bangladesh, 5 Department of Public Health Engineering (DPHE), MOLGRD&C, Dhaka, Bangladesh, 6 WASH Section, UNICEF, Juba, South Sudan, 7 WASH Section, UNICEF, Port Moresby, Papua New Guinea, 8 Department of Civil Engineering, Bangladesh University of Engineering and Technology (BUET), Dhaka, Bangladesh

* mahbubr@icddrb.org

**Editor:** D. Daniel, Gadjah Mada University Faculty of Medicine, Public Health, and Nursing: Universitas Gadjah Mada Fakultas Kedokteran Kesehatan Masyarakat dan Keperawatan, INDONESIA

## Abstract

### Background

The influx of Forcibly Displaced Myanmar Nationals (FDMNs) has left the Southwest coastal district of Cox's Bazar with one of the greatest contemporary humanitarian crises, stressing the existing water, sanitation, and hygiene (WASH) resources and services. This study aimed to assess the existing capacity of local institutions involved in delivering WASH services and identify relevant recommendations for intervention strategies.

### Methods

We used a qualitative approach, including interviews and capacity assessment workshops with institutions engaged in WASH service delivery. We conducted five key informant interviews (KII) with sub-district level officials of the Department of Public Health Engineering (DPHE), Directorate General of Health Services (DGHS), Directorate of Primary Education (DPE) and Bangladesh Rural Advancement Committee (BRAC) to have a general idea of WASH service mechanisms. Seven capacity assessment workshops were organized with the relevant district and sub-district level stakeholders from August 2019 to September 2019. These workshops followed three key areas: i) knowledge of policy, organizational strategy, guidelines, and framework; ii) institutional arrangements for service delivery such as planning, implementation, coordination, monitoring, and reporting; and iii) availability and management of human, financial and infrastructural resources. Data were categorized using thematic content analysis.

**Data Availability Statement:** Due to research ethics (data confidentiality), the data investigated during the current study are not made publicly accessible, however they are available from the corresponding author upon justifiable request. Additionally, ICDDR,B data policy supports the data availability upon request. Request for ICDDR,B research data should be addressed to Ms. Shiblee Sayeed, Senior Manager of Research Administration, at shiblee_s@icddrb.org.

**Funding:** Initials of author who received award: MR Grant number awarded: GR-01809 Full name of funder: United Nations Children's Fund URL: https://www.unicef.org/ Funder's role: YES. Contributed substantially to conception and design of this study.

**Competing interests:** The authors have declared that no competing interests exist.

## Results

The majority of stakeholders lacked awareness of national WASH policies. Furthermore, the top-down planning approaches resulted in activities that were not context-specific, and lack of coordination between multiple institutions compromised the optimal WASH service delivery at the local level. Shortage of human resources in delivering sustainable WASH services, inadequate supervision, and inadequate evaluation of activities also required further improvement, as identified by WASH stakeholders.

## Conclusion

Research evidence suggests that decision-makers, donors, and development partners should consider learning from the WASH implementers and stakeholders about their existing capacity, gaps, and opportunities before planning for any WASH intervention in any particular area.

## Introduction

Despite having the burden of high population density, Bangladesh has attained steady progress in ensuring access to improved safe water and sanitation facilities in the past decade [1, 2]. However, the south-eastern coastal district of Cox's Bazar is one of the most historically under-performing districts in Bangladesh in terms of access to water and sanitation practices and facilities, human development index and child and gender inequities indicators [3]. In August 2017, there was an influx of more than seven hundred thousand people in Cox's Bazar, who fled from Myanmar, and until October 2022, a total number of 943,000 Forcibly Displaced Myanmar Nationals (FDMNs) took refuge around the south-east region of the district, especially in the Ukhiya and Teknaf Upazilas [4, 5]. This resulted in a humanitarian emergency that continues to stress the resources and coping capacity of the local communities and systems until today [6]. This might have affected the developmental progress regarding water, sanitation and hygiene (WASH) service requirements of the overall district of Cox's Bazar [7, 8].

Improving the public health situation by providing WASH service to the population of the relevant Upazilas throughout the district is critical for the WASH Sector at Cox's Bazar. To achieve the goals pertaining to public health, including the preventive and promotion components of health for Universal Health Coverage, appropriate WASH services must be made available in all contexts (such as homes, communities, workplaces, schools and to all people) [9]. However, failure to recognize governance's influence on the availability and accessibility of WASH services and infrastructure hinders progress toward better WASH service delivery [10]. Robust institutions with the capacity to prioritize WASH services are crucial. Assessment of such institutional capacities are required in understanding the dimensions of WASH service coverage for analysing the factors influencing the provision of WASH services [11]. Such assessments could provide an insight into the gap between physical infrastructure and internalization of sanitation and hygiene habits, the quality and quantity of WASH services, the availability of sustainable water supplies, operation and maintenance related challenges, and health-related repercussions of low quality WASH services [11]

In Bangladesh, the Department of Public Health Engineering is the national lead agency for drinking water supply, sanitation and waste management other than the areas in which Water Supply and Sewerage Authority (WASA) operates [12]. WASH services are provided and

maintained in the Cox's Bazar district by DPHE and other Local Government Institutions such as the Health and Family Planning Departments, Primary and Secondary Education Departments, in accordance with their specific areas of responsibility [13] However, these institutions are struggling to provide adequate services due to a lack of robust planning, coordination and feedback mechanism, as well as resource constraints, both human and financial. Additionally, local government, the private sector, WASH service providers, and community leaders lack awareness of the national policies, plans, and frameworks concerning water and sanitation, which further hinders the implementation of WASH service delivery [13]. Hence, we conducted a WASH related institutional capacity assessment of organizations engaged in delivering WASH services in Cox's Bazar district. Findings from this research can not only improve the institutional capacity of organizations in Cox's Bazar but can also inform development strategies in other districts of Bangladesh to enhance the overall WASH service of the country in order to achieve the SDGs target 6.2 and 6.3.

## Methods

### Study setting and participants

Following the discussion with our main collaborators, the Department of Public Health Engineering of Cox's Bazar district and UNICEF (donor), we identified the key institutions and the officials for the assessment considering their involvement in Upazila Water and Sanitation (WATSAN) Committee. A convenient sampling method was used to identify potential participants from those institutions. Participants were contacted by directly visiting or through emails, letters, and phone calls. Efforts were made to engage all relevant institutions considering their organizational hierarchy, distinct principles, and accountability. Upazila-level officials from the respective institutions were invited to participate along with the district-level officials.

### Data collection

A team of anthropologists that included one assistant scientist, a senior research officer and 6 research assistants who have qualitative research experiences performed the qualitative assessment. The assistant scientist led to implementing the overall assessment and all were involved in data collection, analysis and interpretation. Having a team of researchers would minimize the possibility of biasness, and such a team would also be able to handle the high volume of data better.

**Key informant interviews.** We began by conducting five key informant interviews with officials from three government organizations which were DPHE, DGHS and DPE, as well as non-government organizations, to get a preliminary understanding of WASH service mechanisms at the sub-district or Upazila level. These key informants also assisted us in identifying other potential WASH stakeholders from NGOs and other private sectors, as well as guided our approach and key discussion areas of the capacity assessment. KIIs were conducted and audio-recorded after obtaining written informed consent from the participants.

**Workshops.** Then, we conducted seven capacity assessment workshops during August-September 2019, with personnel from other government, non-government and private organizations engaged in WASH-related activities in Cox's Bazar (Table 1). The government organizations included DPHE, DPE, DSHE (Directorate of Secondary and Higher Education), DGHS, DGFP (Directorate General of Family Planning), LGIs such as *Upazila Parishad* and *Union Parishad*. The non-government organizations included were BRAC, Dushtha Shasthya Kendra (DSK), the Village Education Resource Center (VERC), iDE, formerly known as International Development Enterprises) and private organizations included were RFL, Gazi Tanks,

**Table 1. Participants of the capacity assessment workshops and KII.**

| Institutions | Number of participants in the workshop | Number of participants in the KII |
|---|---|---|
| Department of Public Health Engineering | 12 | 2 |
| Directorate General of Health Services | 18 | 1 |
| Directorate General of Family Planning | 17 | 0 |
| Directorate of Primary Education | 15 | 1 |
| Directorate of Secondary & Higher Education | 12 | 0 |
| Union Parishad | 14 | 0 |
| NGOs/Civil Society | 14 | |
| Private sector representatives, dealers, and entrepreneurs | 3 | 1 |

National Polymer and Bengal Plastic. We ensured participation of all designated officials and stakeholders from each Upazila. Nominated alternatives were present in case of the absence of any participants.

The workshops were led by an experienced moderator with knowledge of the local language. A guideline was prepared in Bengali to support facilitation and later translated into English for documentation. The guideline was generated to identify institutional capacity factors, such as: i) knowledge of policy, organizational strategy, guidelines, and framework, ii) institutional arrangements for service delivery such as planning, implementation, coordination, monitoring, and reporting, and iii) availability and management of human, financial and infrastructural resources.

Each workshop consisted of only participants from one institution. This allowed the participants to speak freely and avoid conflicting opinions about their collaborative work. The sessions were audio-recorded with verbal consent from the participants to avoid missing or misinterpretation of data. Moreover, two team members took detailed notes and listed major points on flipcharts. The duration of each workshop was approximately 2–3 hours.

### Data analysis

The team prepared transcripts by summarizing the data from notes and audio recordings shortly after each interview and workshop. A coding framework was developed using the key components of institutional capacity assessment as broad themes and underlying questions as sub-themes which was discussed and agreed upon with the wider study team before analysis. Elements of these broad themes and sub-themes were then extracted from the data by two team members trained in qualitative research methodology. The method of thematic analysis was employed, which helped to identify, analyze, and interpret the patterns of meaning within the qualitative data. Fig 1 shows the institutional capacity assessment data collection and analysis process.

## Results

### Local institutions and programs related to WASH service delivery in Cox's Bazar

DPHE is the national agency for water sanitation and hygiene initiatives and provides services in both urban (except in the four areas where Water Supply and Sewerage Authority supports) and rural areas. DPHE provided WASH infrastructure and services at the district level, such as Cox's Bazar. DPHE also coordinates with DGHS, DGFP, DPE, DSHE, and LGIs) such as *Upazila Parishad* and *Union Parishad*, to support the WATSAN Committees. WATSAN Committees are established by the Government of Bangladesh at all administrative levels—district,

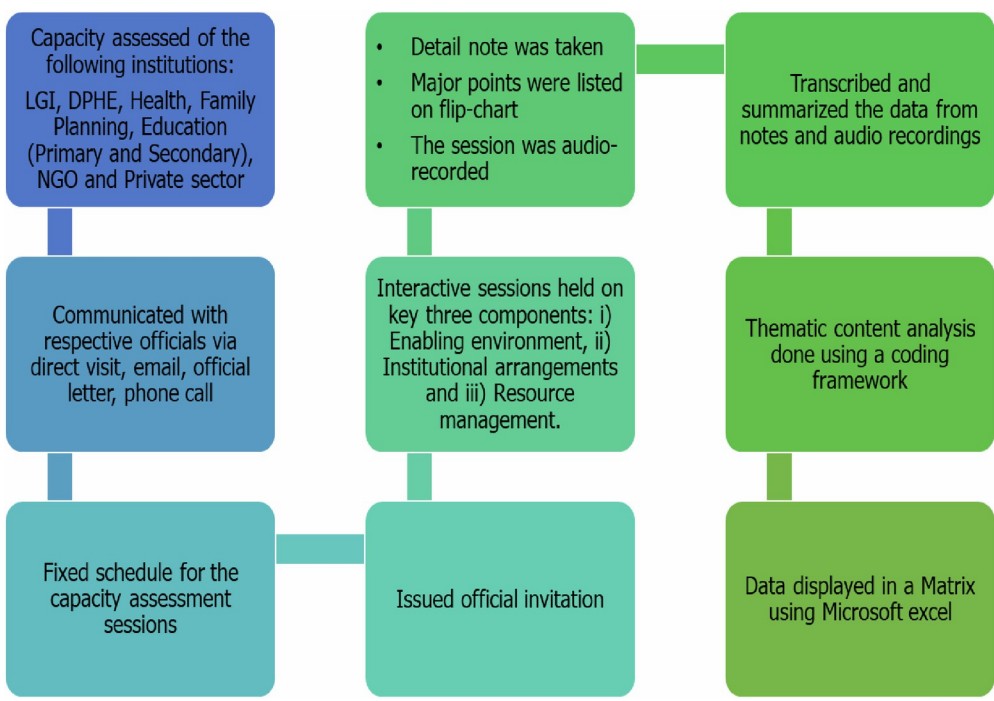

**Fig 1. Flow chart of institutional capacity assessment through workshops.**

sub-district, unions, and wards to ensure safe water supply, sanitation, and development of personal health care for the community people. These committees comprise the respective district, sub-district, union and ward-level officials and representatives from DGHS, DGFP, DPE, DSHE, and other LGIs, as illustrated in Fig 2.

The Union WATSAN committee includes respective frontline workers from DGHS and DGFP, the Headmaster or any teacher of Primary and Secondary Schools, all Union Parishad Members, the Union Parishad Secretary, Tubewell Mechanic of DPHE, a religious leader and an NGO representative (if available). The Union Parishad Chairman acts as the Chairman of that Union WATSAN committee. The Upazila WATSAN Committee and other respective WATSAN Committees coordinate to ensure improved WASH facilities and promotion activities in their jurisdiction. Upazilas or Union Parishads are referred to as a body of people elected to manage the affairs of an Upazila or union level.

Other than the WATSAN committees at different administrative levels, other committees coordinated or managed institutional activities in many ways. Standing and School Management Committees (SMC) are important to mention among them. Standing committees were formed to support the Union Parishad in planning and implementing services on different issues such as education, health, family planning, social welfare and disaster management, agriculture etc., with transparency, accountability, and people's participation. The SMCs engage local people to monitor school management to ensure the development activities of the school. Upazila-level officials from DPE and DSHE guide headteachers and SMCs to ensure safe drinking water for the students. DPHE allocates budget and constructs WASH blocks in Primary Schools under the Primary Education development program-4 (The government's flagship initiative in the education sector for FY2019–FY2023) [14]. WASH blocks are gender-segregated, safely-managed sanitation facilities or latrines with running water sources. In addition, the School Learning Improvement Plan (SLIP) initiative provides modest funds to

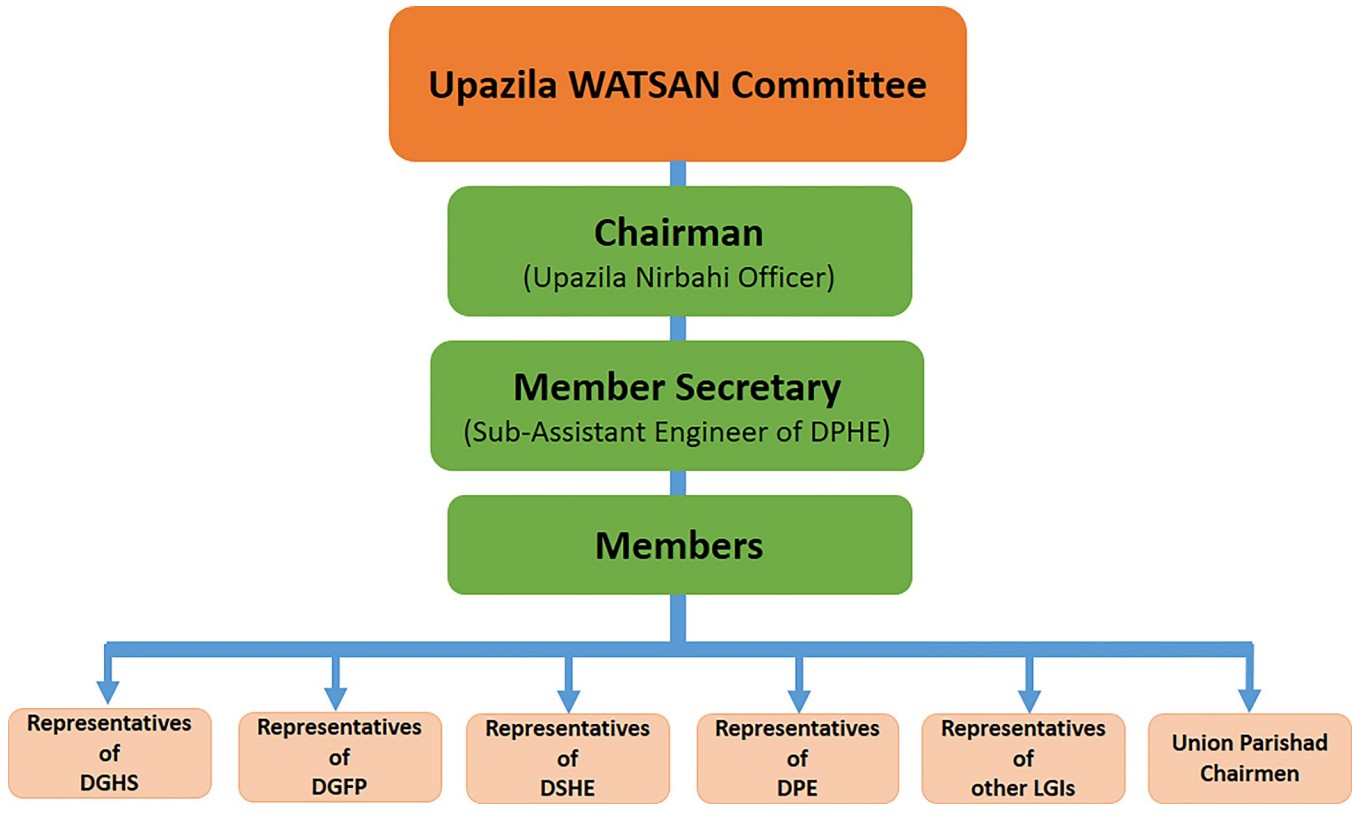

**Fig 2. Structure of Upazila WATSAN committee.**

schools to manage and implement activities according to their own identified needs to improve the quality of education at the school level. DPHE also constructs WASH blocks in cyclone shelters. A cyclone shelter is an elevated building providing security from disasters such as cyclones and associated storm surges. Most cyclone shelters in Bangladesh are elementary schools [15].

In Bangladesh, community clinics and family welfare centers (under DGHS and DGFP, respectively) in rural areas provide primary-level healthcare and family planning services to the community. Although constructing WASH facilities in community clinics and family welfare centers is not linked with DPHE or WATSAN committees, DPHE's responsibility for water supply indirectly affects the construction works at the community clinics and family welfare centers.

NGOs have their priorities and goals of service delivery as per donor requirements, which are time- and area-limited and may differ from the Government Institutions' responsibilities; however, NGOs usually offer their services by co-coordinating and collaborating with LGIs. Private sector representatives, dealers and entrepreneurs manufacture and sell sanitation products based on the choice, customers' budget, or economic status. In some cases, they work jointly with Union Parishad and NGOs, maintaining a private-public partnership to work towards hygiene promotion as a part of their marketing policy. Some NGOs (e.g., BRAC) provide loans to dealers to manufacture specific sanitation products and to sell those products in the open market. The institutional responsibilities of the active organizations in the context of Cox's Bazar are provided in Table 2.

**Table 2. Institutional responsibilities on drinking water supply, sanitation and hygiene.**

| Institutions | Key responsibilities | | |
|---|---|---|---|
| | **Drinking water supply** | **Sanitation** | **Hygiene** |
| Department of Public Health Engineering (DPHE), Local Government Division, MOLGRD&C | ▪ DPHE ensures clean water, establishing iron and arsenic removal plants.<br>▪ DPHE identifies underprivileged populations and unsuccessful areas, where a safe water layer is unavailable.<br>▪ DPHE allocates water sources as per demand and installs them at community and primary schools.<br>▪ Coordinate and maintain liaison with the Upazila Administration, Upazila Parishad, Union Parishads and other relevant government and non-government organizations.<br>▪ Test water quality during installation.<br>▪ Assist community people/education institutions to repair water sources. | ▪ DPHE allocates budget and constructs WASH block in Primary Schools under PEDP (Primary Education development program)-4.<br>▪ Facilitates and organizes sanitation-month observation program with the participation of different organizations. ▪ DPHE has Water Supply and Environmental Sanitation Project (Phase-2) | Facilitates and organizes hand hygiene related programs with the participation of different organizations. |
| Directorate of Primary Education, Ministry of Primary and Mass Education | ▪ Department of Primary Education guides Head Teacher and School Management Committee (SMC) to ensure safe drinking water for the students. | ▪ Monitors construction work of WASH block, toilets, and water supply system in the schools.<br>▪ Provides budget for WASH block repair and maintenance to the schools.<br>▪ Monitors to ensuring the cleanliness of school toilets, WASH blocks and availability of soap and water by the school authority. | ▪ The Department of Primary Education (i.e., Upazila Education Officers and Upazila Assistant Education Officers) guide Head Teacher and SMC about WASH activities along with other hygiene issues.<br>▪ Support to organize day observations events at schools. |
| Directorate of Secondary and Higher Education, Ministry of Primary and Mass Education | ▪ Department of Secondary Education guides Head Teacher and Chairman of School Management Committee (SMC) to ensure safe drinking water for the students. | ▪ Monitor WASH facilities to keep clean, hygienic and functional and conduct feedback session according to the identified situation.<br>▪ Follow-up about the fund-raising of own institution and its use in the cleaning of WASH facilities. | ▪ Ensure disseminating WASH-related message during the assembly session.<br>▪ Ensure cleaning of school premises every Thursday at every school.<br>▪ WASH issues are discussed during meetings of school management committees. |
| Directorate General of Health Services, Ministry of Health and Family Welfare | ▪ Ensure safe drinking water in HCF through Health Engineering Department (HED). | ▪ Ensure sanitation facilities in HCF through Health Engineering Department (HED). | ▪ Sanitation Inspectors motivates community people about hygiene and conduct regular meetings on hygiene issues at the marketplaces.<br>▪ Monitor waste management at marketplaces including slaughterhouse and fish markets. In addition, they monitor food safety and hygiene of food courts and food shops.<br>▪ Health staff are supposed to discuss WASH issues during routine Expanded Program on Immunization (EPI) sessions; however, this is often overlooked due to the workloads of EPI. |
| Directorate General of Family Planning under the Ministry of Health and Family Welfare | | | ▪ Family Planning staff conducts handwashing sessions with the students at secondary schools.<br>▪ Conducts courtyard meetingsabout personal hygiene (one per union per month). |
| Local Government Institutes (Upazila Parishad) | ▪ Upazila Parishad engaged with the Union Parishads and other national organizations under its jurisdiction to coordinate water, sanitation, hygiene, and waste management initiatives. | | |
| Local Government Institutes (WATSAN Committee) | ▪ The union level WATSAN committee is responsible to implement and monitor WASH activities in union level. Union Parishad in collaboration with Upazila Parishad and other national agencies is mandated to coordinate those activities. | | |

(*Continued*)

**Table 2.** (Continued)

| Institutions | Key responsibilities | | |
|---|---|---|---|
| | **Drinking water supply** | **Sanitation** | **Hygiene** |
| NGOs and Private Sectors | ▪ Different NGOs in Cox's Bazar works in Rohongya camps and host communities to support water, sanitation, and hygiene activities. This includes free materials for construction of latrine, emergency WASH services, hygiene campaigns, menstrual hygiene management support etc.<br>▪ Different NGOs and private sector program offers different product variations of the latrine solution: i) Sato Pan, ii) a slab for a direct pit (SanBox), iii) the pan and a slab for an offset pit (Sanbox). NGOs train local entrepreneurs to meet demand for sanitation products. One NGO (IDE) also created Sato pan upon teaming up with RFL Plastics Ltd (a private company). | | |

## WASH conditions in the context of Cox's Bazar

Participants reported inadequate WASH practices in public areas, marketplaces and community clinics. For example, a few participants expressed that people are less aware of personal hygiene, and some do not use soap for regular handwashing due to poor economic conditions. For instance, in one union, participants from the LGI reported that half of the population used soap after using latrines, and some did not practice handwashing at five critical times during the day. About fecal sludge management, DGFP participants mentioned that many latrines overflow during floods and people practice manual emptying and open disposal of fecal sludge, which causes an unhealthy environment for all. Participants from both DPHE and DGFP reported that unclean toilets and the lack of gender-segregated toilets in cyclone shelters prevented many people from using them. This maintenance problem is also revealed in one of the Bangladesh Government projects (Ashrayan Prokolpo), where disadvantaged poor people from different places take shelter. Mentioning the situation from a particular Upazila, one key informant stated that each Ashrayan project accommodates about 120 families, with two toilets for every six families and separate bathing facilities. However, the users struggle with proper maintenance and cleaning of the toilets. As a result, the toilets become unusable quickly. Lack of proper coordination and appropriate budgeting for maintenance, inadequate design provisions against disaster (e.g., placing the WASH facility below flood level), lack of consideration for water quantity and quality (e.g., unavailability of running water, absence of Iron Removal Plant) and poor construction materials have rendered most of the WASH blocks damaged or unusable, as reported by the Primary School Officials.

Participants from DGHS and DGFP mentioned that they face problems in maintaining and renovating WASH facilities at the many community clinics and family welfare centers because they do not have a running water supply and electricity. DGFP participants mentioned that about 20% of their family welfare centers have no running water supply. Also, there is no maintenance plan for WASH facilities at the family welfare centers and due to the lack of proper cleaning and maintenance, DGFP own study findings showed that nearly 40–50% of facilities are unhygienic. DGHS participants mentioned that approximately 90% of tubewells at the community clinics are not functional. They added that they do not get water from the tubewells in the hilly areas since the groundwater levels have declined.

DPE participants mentioned that many WASH blocks in Pekua, Kutubdia and Chakaria Upazila constructed during 2013–2015 were not functional due to the unavailability of electricity and running water supply. High concentration of iron in the water also leaves a stain in the sanitary ware, damaging the WASH blocks. They also mentioned that consideration of the flood level for constructing WASH blocks is important and should ensure good quality construction materials for WASH blocks for the longevity of those facilities. Lack of menstrual hygiene management (MHM) facilities at school toilets may have been a reason behind the absence of female students during menstruation, as attributed by DSHE participants.

Private sector participants mentioned that there is inadequate availability of need-based sanitary ware or products that are appropriate for different geographical contexts. Also, people are unaware of quality or sustainable products; for example, plastic slabs are less durable than ceramic slabs, but considering cost, people buy less sustainable products. Therefore, local entrepreneurs and private sectors may collaborate to ensure quality products at a low cost.

## Knowledge of policy, organizational strategy, guidelines, and framework

Bangladesh has several national policies, plans and strategies related to water supply, sanitation, and hygiene. Our findings revealed that most participants had not heard about those policies and were unaware of their roles and responsibilities. For instance, one participant explained the policy in the following manner

> *'Policies include ensuring clean water, managing advanced sanitation system, plant for removing arsenic and iron from water.'*

Another participant explained that they identify poor and unprivileged people through locally elected members and then provide latrines or tube wells. However, he remained unsure about any clear guidelines regarding this. Representatives from other government institutions (DGHS, DGFP, DPE, DSHE and LGIs) also could not mention the institutional policy, strategy, organizational mandates, and regulatory framework that support delivering WASH services at their level. However, they all acknowledged the necessity of training in this regard.

Although private sector participants mentioned receiving 'hygienic latrine' training from NGOs, most of them never received any training or guidelines from the government on other WASH issues such as iron and other pollutant-free safe drinkable water and sustainable and environment-friendly sanitation products.

## Inadequate consideration of bottom-up feedback

DPHE local teams implement programs after consultation with Upazila Executive Officer and local government institutions as per their allocation received from the DPHE Head Office. Every year Union Parishad chairmen collect information through their respective Union Parishad Members at the ward-level WATSAN committee. Similarly, the Upazila WATSAN committee collects information from all Union Parishad Chairmen, informs the district WATSAN committee, and later gives the compiled demand to DPHE, but when DPHE allocates, it does not match the bottom-up recommendation. Besides, the allocation procedure deprives marginalized people of becoming beneficiaries of the intervention. A key informant mentioned:

> *'We usually receive the allocation for construction of tubewells in this Upazila 3 times a year from the 'Polly Onchole Pani Sorboraho Prokolpo' (Village Water Supply Project) and is distributed through the Upazila WATSAN committee. 50% of this allocation is received by the Upazila Chairman and 50% by the local Member of Parliament (MP).'*

The participant mentioned that the budgeting system must be changed, adding that:

> *'Around 40% should be decided by the MPs, 40% by the WATSAN committee members, and the rest 20% should be kept preserved for the demand from the institutions itself, to decide'.*

Also, this type of top-down approach suggested for the budgeting system is not always context-specific. For example, different areas have different requirements. In certain areas of the

district, decreasing groundwater levels and high concentrations of iron and salinity in surface and groundwater were reported as key challenges. Another key informant mentioned that there are some areas with a high concentration of iron, stating that:

*'If an iron treatment plant or water plant can be set up in different union parishads and piped networking system can be ensured for water supply, that will be an effective step. Also, purified surface water should be used, and constant running water sources should be closed to stop wasting natural sources and avoid water scarcity in the future.'*

## Insufficient coordination between organizations

There is absence of effective coordination among DPHE and other members of the WATSAN committee. Meetings are not arranged regularly. Although Upazila WATSAN committee meetings are supposed to be organized once every two months, they are arranged only in cases of emergencies and priority issues. One key informant mentioned that yearly only one or two meetings of the WATSAN committees are organized. Since the members of this committee are regularly connected for work purposes, he thinks the work is usually done properly even if there is a lack of regular meetings held by the WATSAN committee.

We found inadequate coordination between the Upazila and Union WATSAN Committees or Union Standing Committee. In a workshop, a few participants reported that,

*'Lack of coordination between WATSAN and standing committees hamper taking effective measures for WASH facilities. To make it more effective at the Upazila and union levels, a few members could form a new committee that is only responsible for monitoring WASH services. Since the WATSAN committee is mainly responsible for WASH, this committee should be activated to make it more accountable and effective in carrying out coordination meetings. Monthly meetings need to be arranged at both Upazila and union levels.'*

Participants mentioned that the lack of inspection for waste management in marketplaces, such as fish markets and slaughterhouses, makes these places unclean and unhygienic. Participants also reported that the lack of coordination between the Upazila WATSAN committee and SMC has affected the implementation of WASH activities at secondary education institutes.

The local Government has no separate or specific WASH reporting and monitoring system for Union Parishad. Despite the fact that Union Parishad and DPHE work collaboratively, there are no operations or monitoring systems that are systematically designed to ensure the sustainability of infrastructure.

Since there are no systematic arrangements for quality assessment from the government or NGOs, some local traders or entrepreneurs profit from selling low-quality sanitary ware or products (e.g., toilets and basins). The entrepreneurs recommended training, provision of loans with low-interest rates, and quality assurance mechanisms for attaining quality WASH products in the market.

## Inadequate resources (human resources, financial resources and infrastructural resources) hampered WASH service delivery

DPHE has representation at the national and sub-national levels to ensure WASH services. However, DPHE participants mentioned that due to the high influx of Rohingya refugees in Cox's Bazar, there is a shortage of manpower to work in the host community of Ukhiya and

Teknaf Upazila. In the district, optimal service delivery is hampered due to less skilled human resources and inadequate financial resources. One participant mentioned that to increase the skills of the mechanics, they need to be trained on issues such as appropriate places for tube wells to be installed, the water level, suitable technologies etc. Inadequate financial resources for transport costs and technical logistics supply are other constraints for optimal WASH service delivery. DPHE receives the same amount of allocation for transport allowance of their staff for all Upazilas. This amount is often inadequate for larger Upazilas, such as Chakaria having 18 unions.

DPHE mentioned lack of transport support sometimes hampered their regular activities and made them unable to render the services at remote unions on time. They mentioned that they can make a specific number of limited visits in a year to a remote area due to a limited budget, and for these visits, they do not get any vehicle support. One key informant mentioned,

> 'vehicles' should be arranged for the mechanics of this office (Upazila DPHE). There is an allowance of BDT 5,000 for such travel, which is divided among four mechanics. This allocation is insufficient, so often these mechanics must go to different places at their own expense, making it impossible for them to work well'.

Another key informant revealed that in some remote areas, such as in Ramu Upazilla, deep-layer digging is required for deep tubewells. However, the DPHE cannot provide the appropriate technology, such as submersible pumps or drilling equipment, to install such tubewells due to the absence of electricity in areas such as *Gorjonia*. Sometimes they can manage the electricity through generators, which results in high costs. In addition, some areas of Teknaf and Ukhia Upazila of Cox's Bazar have an excessive rocky layer where installing tubewells is inappropriate, and alternative ways, such as pipelined water supply system, need huge investments and bigger plans. DPHE participants also mentioned that rocky layers make it difficult to install deep tube wells, and there is not enough budget for cutting or drilling through those layers. One participant from DPHE mentioned that the budget allocation should be increased for the hilly areas according to the rate of hilly areas. Regarding implementation challenges by the private contractors and mechanics, one key informant mentioned that,

> 'Contractors and mechanics are not government staff but are rather 'private' workers. They often do not want to go to work in some 'hard-to-reach' areas like Gorjonia and Kocchopia. To hire them, they need to pay some extra 'tips', which also raise the overall cost of the installation'.

In general, budgets are prepared using a top-down approach. Local officials do not have any role or engagement in budget preparation and planning. In the workshops, participants could not inform anything about the budget allocation of the current fiscal year. Participants also reported not having a specific budget allocation to implement WASH-specific activities (e.g., for providing inclusive wash blocks in secondary education institutes).

Both DPE and DSHE have an adequate workforce at the Upazila level, and they visit the schools following their monitoring protocols. DPE participants mentioned they have an adequate workforce to perform WASH activities. Upazila Education Officers regularly visit the schools and inform the principal regarding the maintenance and usage of toilets. Besides, they sent a proposal for the new construction of required WASH blocks in the school with the help of DPHE.

In addition, participants from the DGHS mentioned having a separate workforce to ensure the quality of WASH activities/services and maintenance of WASH facilities in the healthcare centers. But there is no exclusive staff for WASH service delivery. Frontline workers/community health workers do activities related to issues around WASH service and practices at the community level (e. g. courtyard session, one-to-one session etc).

Similarly, frontline workers of DGFP perform some awareness-raising works related to hand washing and personal hygiene. Participants from DGFP mentioned having an allocation of only TK 700 per year to maintain the cleanliness of each Family Welfare Centre.

## Excess workload

WASH counseling activities are often compromised due to the existing workload of the frontline workers of DGHS caused by the large population of FDMNs in the area. The health centers, such as community clinics, lack functional WASH facilities and maintenance capacity, resulting in inadequate WASH practices by both health service providers and patients or caregivers. However, one key informant mentioned that Upazila Health Complex (Upazila level health facility) arranges different programs in schools, such as "National Worm Control Week (*krimisoptaho*)" and "Jatiyo Sastho Seba Soptaho" (National Health Service Week), where the Health Assistants spread awareness among the students regarding hygiene practices. Handwashing campaigns are also organized in the "Little Doctors *(Khude Doctor)*" program, where the students are instructed to demonstrate handwashing properly. This creates an opportunity for the Upazila Health Complex to raise awareness among people about WASH.

## Recommendations from DPHE

DPHE participants recommended including disabled or physically challenged people while planning or budgeting for any WASH facility construction. They also recommend biofil-toilet technology to ensure safely managed sanitation facilities at the cyclone shelters. Biofil toilets have an environmentally friendly digester system, which treats organic waste through aerobic decomposition. The arrangement of separate toilets for the disabled and physically challenged people at the cyclone shelters was also another recommendation. They mentioned that the Union WATSAN and SMC meetings should be regularized, and the committee should inform the decisions to the Upazila DPHE. Moreover, the WATSAN committee should prepare monthly reports.

DPE participants recommended determining water extraction techniques and installation design based on local realities. They also recommended using surface water to ensure a pipeline water supply network to each school or group of schools. They mentioned the requirement of a concise and integrated policy and strategic guideline for conducting WASH activities at the local level that will include health, education, and WASH together.

DPE and DSHE participants recommended that MHM be considered during the construction of WASH blocks in schools and for cyclone shelters. Emergency sanitary napkins can be promoted with the support of the SLIP fund, as recommended by one DPE participant. Teachers, SMC, and scout groups could be trained to raise awareness of the WASH needs and requirements of the students.

## Recommendations from the private sectors

Private sector participants recommended building awareness among the community people about sustainable WASH products at the household level for market promotion and collaborating with local NGOs and Union Parishad. They also mentioned that relevant institutions

might collaboratively develop and share the context-specific plan with the private sector. The private sector or entrepreneurs can follow that plan within their capacity.

## Discussion

The study may not fully reflect other areas of the country which have not experienced a similar level of investment. Nevertheless, the current findings revealed certain areas of improvement expressed by WASH stakeholders which should be considered by decision-makers, donors, and development partners for improving the performance of effective WASH service delivery. Recently, an increasing number of international and local organizations have taken an interest in strengthening WASH services in the overall host community of Cox's Bazar, and not only to the FDMNs [16, 17]. This analysis helps the policymakers to understand their key strengths and weaknesses to allow for targeted interventions and sustainable progress in the future. It is also essential to identify the key stakeholders and explore and understand their roles and capacity in service delivery and policy decisions [18]. Our study provided evidence supporting close engagement with public providers who are involved in service delivery [19] and has helped to understand how existing WASH interventions work in "real world" settings [20].

In general, all institutions have an average capacity to deliver WASH services in their specific jurisdiction. Although, other institutions did not have any dedicated program for WASH service delivery, except DPHE. The findings indicate a lack of knowledge of the different stakeholders of legal and organizational strategy, guidelines, and framework, specifically the institutions not directly involved in providing WASH services at the community level. The results of this study strongly suggest the specific raising of awareness for all relevant staff/officials of each relevant institution on WASH, technologies related to safely managed water and sanitation facilities, SDG targets, and WASH policies.

A strength of our study lies in the fact that we spoke to WASH implementers, relevant shareholders, institutional managers, policymakers, and the private sector, even though their potential roles are often ignored by the (WASH) development sector [21]. It's crucial to comprehend how the private sector may contribute to the development of water and sanitation infrastructure as well as increased water system efficiency. WASH stakeholders are now encouraged to engage the private sector more for achieving the Sustainable Development Goal 6 (Clean water and sanitation), which may pave the way for improvement [22]. The capacity of the private sector should be enhanced, especially on diversified technologies and promotion of WASH products, considering their role in the sustainable improvement of the WASH market.

Many institutions within the private sector had limited capacity to deliver WASH services, and they do not perceive WASH as their prioritized responsibility. In addition, the ineffective implementation of a bottom-up work plan is another challenge due to the top-down budgetary system. Despite the plans being developed with the active participation of local-level elected members and chairmen, while implementing these plans, political interference and insufficient funding often hindered the projected outcome. A study on governing WASH in Malawi also reports that WASH governance and service delivery are significantly impacted by insufficient financial resources and a lack of organised funding systems [10]. Therefore, our suggestion is to promote bottom-up, participatory and long-term plans at Union and Upazila levels, and these strategies should be developed by the lead role of DPHE in coordination with the local government, education, health and private sector. This plan should include installation of WASH facilities and promotion of WASH practices at all levels.

We identified a deficit of effective coordination among the key WASH stakeholders; WATSAN committee meetings were not held on time. Coordination and collaboration are required to help the WASH stakeholders achieve quality decision-making, specifically policy

formulation and implementation. It will help the WASH stakeholders identify their mutual interests and increase the opportunity for working together to facilitate investment planning in the short- to long-term [23]. The study suggests activating the WATSAN committee at all administrative levels with regular meetings and establishing joint monitoring mechanisms and report-sharing at the local level [24]. Private Sector representatives could be co-opted on the WATSAN committees.

Sustainable WASH service delivery in a particular area requires the planned allocation of financial and human resources [25]. We found that many of these institutes engaged in WASH services delivery lack dedicated funds or workforce to deliver WASH services in Cox's Bazar [26]. We identified a lack of knowledge of context-specific problems and, thus, the implementation of top-down, insufficient financing as a barrier to service delivery [27]. Having an explicit understanding of the context is essential for planning implementation research in a particular sector [28]. Similar arguments were made in a study on South Africa where insufficient allocation of finances into WASH resources due to a lack of political will at the governmental levels have resulted in poor implementation of plans around water safety [29]. Hence, to achieve safely managed/advanced water and sanitation, an upazila-level implementation guideline that takes into account the context is necessary. WASH service delivery can be accelerated through a range of interventions beyond training, including policy alignment and strengthening of monitoring systems and coordination mechanisms through existing intersectoral platforms.

## Implications for research and policy

Capacity assessment helps to improve the capabilities of individuals and organizations to function efficiently to attain sustainable results [30]. Our findings can help service providers or implementers to get evidence of strengths and weaknesses and the private sector to focus on demand-based production and marketing of contextually appropriate WASH products/technologies. In combination, these findings will help all to improve the current condition in WASH service delivery in a particular area by improving the existing capacities of the institutes and coordination among the relevant stakeholders. Strengthening institutional capacity collaboratively will improve several areas of service delivery [31–33]. The assessment of institutional capacity shows the gaps in delivering WASH services and opportunities to improve the WASH sector at the district level. Engaging the private sector and researchers in the WASH sector networks is a prerequisite for continuous improvement. Further studies and periodic monitoring and review of the WASH situation should be done to evaluate the effectiveness of interventions in achieving the SDG targets.

## Conclusion

Institutional capacity assessment and sustainable improvement of the existing WASH program is an integrated approach for effective WASH service delivery. This paper generates evidence for the scope of required improvement that could help the entire WASH sector of Cox's Bazar district to gear them up to reach the SDG target of achieving access to adequate and equitable sanitation and hygiene for all by 2030. Key stakeholders recognised that if plans could be developed at the local level, then the implementation strategy would be more context-specific and appropriate. Average milestones were set to achieve the goal, although the respective Upazila team determined their milestone considering their WASH situation and local context. The research evidence suggests that decision-makers, donors, and development partners should consider learning from the WASH implementers and stakeholders about their existing capacity, gaps, and opportunities before planning for any WASH intervention in any particular

area. This strategy could also be implemented in other low-and-middle-income countries to reach the SDG target by 2030.

## Limitations

After initial three workshops, we experienced difficulties in assembling all of the Union Parishad Chairmen on a particular day, due to busy schedules around various working agendas. In addition, interruption in the actual schedule caused dropouts of other participants/officials as well. Moreover, it was often difficult to retain them for such a long period of time. Therefore, strategy of participatory planning was modified to reach out to all respective Union Parishad Chairmen from those unions which were not included in the sample of qualitative data collection of this current study. In addition, assigned team collected the required information such as, availability, accessibility and requisite for safe water, improved sanitation and hygiene services at their corresponding communities and institutions, prior to the day of workshop.

## Acknowledgments

The authors are grateful to the study participants for their valuable time, and team members who were involved in data collection. The authors also acknowledge with gratitude the commitment of UNICEF and the Department of Public Health Engineering (DPHE) to its research efforts. icddr,b is thankful to the Governments of Bangladesh and Canada for providing their core and committed support.

## Author Contributions

**Conceptualization:** Mahbubur Rahman, Mahbub-Ul Alam, Sharmin Khan Luies, Zahidul Mamun, Musarrat Jabeen Rahman, Tazrina Ananya, Dara Johnston, Martin Worth, Umme Farwa Daisy, Tanvir Ahmed.

**Data curation:** Sharmin Khan Luies,  Asadullah, Abul Kamal, Umme Farwa Daisy.

**Formal analysis:** Sharmin Khan Luies,  Asadullah, Abul Kamal, Umme Farwa Daisy.

**Funding acquisition:** Mahbubur Rahman.

**Investigation:** Mahbubur Rahman, Mahbub-Ul Alam, Sharmin Khan Luies, Sharika Ferdous, Debashish Biswas, Tazrina Ananya, Abul Kamal, Umme Farwa Daisy, Tanvir Ahmed.

**Methodology:** Mahbubur Rahman, Mahbub-Ul Alam, Sharmin Khan Luies, Zahidul Mamun, Debashish Biswas, Tazrina Ananya,  Asadullah, Abul Kamal, Dara Johnston, Martin Worth, Umme Farwa Daisy, Tanvir Ahmed.

**Project administration:** Mahbubur Rahman, Mahbub-Ul Alam, Sharmin Khan Luies, Sharika Ferdous, Debashish Biswas, Tazrina Ananya,  Asadullah, Abul Kamal, Umme Farwa Daisy, Tanvir Ahmed.

**Resources:** Mahbubur Rahman, Mahbub-Ul Alam, Zahidul Mamun, Tanvir Ahmed.

**Supervision:** Mahbubur Rahman, Mahbub-Ul Alam, Debashish Biswas, Tanvir Ahmed.

**Validation:** Mahbubur Rahman, Mahbub-Ul Alam, Sharmin Khan Luies, Sharika Ferdous, Debashish Biswas, Umme Farwa Daisy, Tanvir Ahmed.

**Visualization:** Sharmin Khan Luies, Sharika Ferdous, Musarrat Jabeen Rahman, Abul Kamal, Umme Farwa Daisy, Tanvir Ahmed.

**Writing – original draft:** Mahbubur Rahman, Mahbub-Ul Alam, Sharmin Khan Luies, Musarrat Jabeen Rahman, Debashish Biswas, Tazrina Ananya,  Asadullah, Abul Kamal, Dara Johnston, Martin Worth, Umme Farwa Daisy, Tanvir Ahmed.

**Writing – review & editing:** Mahbubur Rahman, Mahbub-Ul Alam, Sharmin Khan Luies, Sharika Ferdous, Zahidul Mamun, Musarrat Jabeen Rahman, Debashish Biswas, Tazrina Ananya,  Asadullah, Abul Kamal, Ritthick Chowdhury, Eheteshamul Russel Khan, Dara Johnston, Martin Worth, Umme Farwa Daisy, Tanvir Ahmed.

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
