## [Decision Letter · Decision Letter 0]

25 Aug 2023

PONE-D-23-19851Institutional Capacity Assessment in the lens of implementation research: Capacity of the local institutions in delivering effective WASH services at Cox's Bazar district, BangladeshPLOS ONE

Dear Dr. Rahman,

Thank you for submitting your manuscript to PLOS ONE. After careful consideration, we feel that it has merit but does not fully meet PLOS ONE’s publication criteria as it currently stands. Therefore, we invite you to submit a revised version of the manuscript that addresses the points raised during the review process.

Please see editor comments below.

We look forward to receiving your revised manuscript.

Kind regards,

D. Daniel, Ph.D.

Academic Editor

PLOS ONE

Journal Requirements:

5. Please ensure that you refer to Figures 1 to 2 in your text as, if accepted, production will need this reference to link the reader to the figure.

**Additional Editor Comments:**

Please revise your draft according to the reviewers’ (see attachment if any) and editor’s comments.

• I don’t really understand what you mean by “effective” in the title. Please explain it somewhere in the text. I can’t find the exact meaning of this.

• I think the keywords are too much. Please reduce to a maximum of ~5

• Please correct some reference errors.

• There is no space between words and citations in some locations, e.g., lines 478, 482, 484. Please correct the whole draft.

• I think you don’t need a list of abbreviations.

• For the declarations section, please see examples from other publications in PLOS One and adjust your draft accordingly.

• I think the implications are still local. Please expand your implications to other areas as well.

• You can add cite studies from other regions/countries about WASH institutions/stakeholders/governance to support your argument/compare your findings, e.g., https://doi.org/10.3390/w13030314

• I don’t see the conclusion section. Please add it.

Reviewers' comments:

Reviewer's Responses to Questions

**Comments to the Author**

1. Is the manuscript technically sound, and do the data support the conclusions?

Reviewer #1: No

Reviewer #2: Yes

2. Has the statistical analysis been performed appropriately and rigorously? 

Reviewer #1: N/A

Reviewer #2: N/A

3. Have the authors made all data underlying the findings in their manuscript fully available?

Reviewer #1: No

Reviewer #2: Yes

4. Is the manuscript presented in an intelligible fashion and written in standard English?

Reviewer #1: No

Reviewer #2: Yes

5. Review Comments to the Author

Reviewer #1: In the introduction there is reference made to the displaced persons and the potential impact on the resources, however the scope of the document is on the "host community"/Bangladesh population. In many areas the Forcibly Displaced Myanmar Nationals (FDMNs) represent the majority of the population. Specifically in the most water scare parts of the district. However, FDMN/refugees and the service delivery approaches to providing WASH services in the camps is ignored or excluded from the scope. This is ok, but it needs to be better defined 1) why Cox's was selected (vs other districts in BGD) and 2) why only focusing on host population is rigorous/insightful.

In the methods section there is reference to a similar assessment "Institutional mapping and analysis of WASH Services and Costs, WASHCost-CESS Working Paper No", however that approach also included a comparison between the institutional arrangements and service delivery models to quantitative data collected through case studies. It is unclear that without this type of additional, complementary data that the analysis carried out by the authors is very insightful at all. The authors mention in the last sentence of the manuscript that "In addition, assigned team collected the required information such as availability, accessibility and requisite for safe water, improved sanitation and hygiene services at their corresponding communities and institutions, prior to the day of workshop" However this information was not presented anywhere in the manuscript.

Additional comments below and in the attached:

In the findings section there are instances when statistics are provided without reference. “DGHS participants mentioned that 90% of tubewells at the community clinics are not functional.” Specific sources should be provided in all cases where statistics are referenced.

I recommend that the document is restructured to move the information on the overall institutional arrangements to the introduction- or perhaps develop a background section. Much of the information presented in the “results” section is not specific to Cox’s Bazar and not linked to the data collected through the KIIs or workshops.

solid waste is introduced on page 18 for the first time, but it is not included in the table 2. The table should be adjusted or the references to solid waste removed.

“Recommendations from the participants” section is poorly organized. It isn’t clear who should be taking forward any specific recommended action, or if there is any general recommendation on changes to the institutional roles and responsibilities that might require a policy or legislative reforms.

Reviewer #2: I take this opportunity to thank the authors for preparing this useful report. I have a few suggestions to share

Introduction : Page 5 line 76 the abbreviation DPHE needs to be put in full. In this section it would be useful to highlight more the WASH situation in that region from published literature. I suggest you also highlight how the private sector and other humanitarian organisations are involved WASH in this region seeing you mention them as lacking awareness on WASH policies.

You mention the DPHEs roles, do these involve formulation and awareness creation of the national policies, plans, and frameworks concerning water and sanitation? I think it would be useful background information for the reader to understand.

Methods

This section is clearly written. A few comments;

The authors reports that a convenient sampling method was used to identify institutions fully or partially involved in WASH-related activities in Cox's Bazar district. I propose a bit more information on how the partial or full involvement was assessed.

I suggest the authors correct the referencing errors on page 8 line 152 and 154

Results

In this section is propose the authors set up this into two main sections;

One section to describe the setting/context and another to describe the key themes/ findings from the research activities. As it is its challenging to follow at what point the findings of the interviews begin to be described.

Table 2 a very useful table that describe the key actors. However, its is a rather long table, i suggest some edits to the table to make it easier for the reader to follow.

I suggest the authors correct the referencing errors on page 8 line 178 and 179

The main results and thematic areas are well described, i am curious to find out if there were any positive findings or reports by the participants from the interviews and workshops. As it stands the authors highlight only the challenges.

Discussion

This section is well written. I propose the authors, set up the discussion to highlight how their findings sit in light of other assessments that have been done in this area or similar settings. Are there things that are unique to this setting ?

I also propose the authors describe /suggest how these WASH policies can be better developed and utilised in such settings seeing there are different actors with different roles.

Include all the abbreviations used in the report in the list of abbreviations

A minor correction on the figures , both are labelled figure 1 and also these figures were not described/highlighted in the main text

6. PLOS authors have the option to publish the peer review history of their article (what does this mean?). If published, this will include your full peer review and any attached files.

Reviewer #1: No

Reviewer #2: No

---

## [Author Response · Author response to Decision Letter 0]

12 Oct 2023

September 29, 2023

To: Editor, PLOS ONE

Re: Revision and responses to reviewer’s comments

Dear Editor,

Thank you for reviewing our manuscript entitled “Institutional Capacity Assessment in the lens of implementation research: Capacity of the local institutions in delivering WASH services at Cox's Bazar district, Bangladesh” for consideration for publication in the PLOS ONE.

We have made and highlighted revisions by using track changes mode to the manuscript based on editor’s and reviewer’s comments. Here are our point-by-point responses to the comments for kind consideration.

Journal Requirements:

Response: Submission was done according to PLOS ONE’s style requirements. 

Response: icddr,b data policy supports the data availability upon request. Request for icddr,b research data should be addressed to Ms. Shiblee Sayeed, Senior Manager, Research Administration at shiblee_s@icddrb.org.

Response: icddr,b data policy supports the data availability upon request. Request for icddr,b research data should be addressed to t to Ms. Shiblee Sayeed, Senior Manager, Research Administration at shiblee_s@icddrb.org

Response: Thanks for the suggestion, edited in the manuscript. 

5. Please ensure that you refer to Figures 1 to 2 in your text as, if accepted, production will need this reference to link the reader to the figure.

Response: Thanks for the suggestion, edited in the manuscript. 

 

Additional Editor Comments:

Please revise your draft according to the reviewers’ (see attachment if any) and editor’s comments.

• I don’t really understand what you mean by “effective” in the title. Please explain it somewhere in the text. I can’t find the exact meaning of this.

Response: We have dropped effective from the title and revised it. 

• I think the keywords are too much. Please reduce to a maximum of ~5

Response: Reduced to 5 key words

• Please correct some reference errors.

Response: Thanks for the suggestions, we have rechecked and confirmed the reference section. 

• There is no space between words and citations in some locations, e.g., lines 478, 482, 484. Please correct the whole draft.

Response: Corrected across the whole draft. 

• I think you don’t need a list of abbreviations.

Response: Omitted. 

• For the declarations section, please see examples from other publications in PLOS One and adjust your draft accordingly.

Response: Declarations have been written in accordance with other PLOS One publications. 

• I think the implications are still local. Please expand your implications to other areas as well.

Response: We have added a conclusion section where we have highlighted the implications:

“Institutional capacity assessment and sustainable improvement of the existing WASH program is an integrated approach for effective WASH service delivery. This paper generates evidence for the scope of required improvement that could help the entire WASH sector of Cox's Bazar district to gear them up to reach the SDG target of achieving access to adequate and equitable sanitation and hygiene for all by 2030. Key stakeholders recognised that if plans could be developed at the local level, then the implementation strategy would be more context-specific and appropriate. Average milestones were set to achieve the goal, although the respective Upazila team determined their milestone considering their WASH situation and local context. The research evidence suggests that decision-makers, donors, and development partners should consider learning from the WASH implementers and stakeholders about their existing capacity, gaps, and opportunities before planning for any WASH intervention in any particular area. This strategy could also be implemented in other low-and-middle-income countries to reach the SDG target by 2030.”

• You can add cite studies from other regions/countries about WASH institutions/stakeholders/governance to support your argument/compare your findings, e.g., https://doi.org/10.3390/w13030314

Response: Added a few that were found to be relevant. 

• I don’t see the conclusion section. Please add it.

Response: Added. 

 

Reviewers' comments:

Reviewer's Responses to Questions

Comments to the Author

1. Is the manuscript technically sound, and do the data support the conclusions?

Reviewer #1: No

Reviewer #2: Yes

2. Has the statistical analysis been performed appropriately and rigorously?

Reviewer #1: N/A

Reviewer #2: N/A

3. Have the authors made all data underlying the findings in their manuscript fully available?

Reviewer #1: No

Reviewer #2: Yes

4. Is the manuscript presented in an intelligible fashion and written in standard English?

Reviewer #1: No

Reviewer #2: Yes

5. Review Comments to the Author

Reviewer #1: 

In the introduction there is reference made to the displaced persons and the potential impact on the resources, however the scope of the document is on the "host community"/Bangladesh population. In many areas the Forcibly Displaced Myanmar Nationals (FDMNs) represent the majority of the population. Specifically in the most water scare parts of the district. However, FDMN/refugees and the service delivery approaches to providing WASH services in the camps is ignored or excluded from the scope. This is ok, but it needs to be better defined 1) why Cox's was selected (vs other districts in BGD) and 2) why only focusing on host population is rigorous/insightful.

In the methods section there is reference to a similar assessment "Institutional mapping and analysis of WASH Services and Costs, WASHCost-CESS Working Paper No", however that approach also included a comparison between the institutional arrangements and service delivery models to quantitative data collected through case studies. It is unclear that without this type of additional, complementary data that the analysis carried out by the authors is very insightful at all. The authors mention in the last sentence of the manuscript that "In addition, assigned team collected the required information such as availability, accessibility and requisite for safe water, improved sanitation and hygiene services at their corresponding communities and institutions, prior to the day of workshop" However this information was not presented anywhere in the manuscript.

Response: Thank you so much for this comment. There was plenty of research conducted on the Rohingya refugee population inside the camp context and some of those papers already described the capacity of the local authority, development partners and local agency in terms of service delivery. Here are some examples:

Alam M-U, Unicomb L, Ahasan SMM, Amin N, Biswas D, Ferdous S, Afrin A, Sarker S, Rahman M. Barriers and Enabling Factors for Central and Household Level Water Treatment in a Refugee Setting: A Mixed-Method Study among Rohingyas in Cox’s Bazar, Bangladesh. Water. 2020; 12(11):3149. https://doi.org/10.3390/w12113149)

https://gbvguidelines.org/wp/wp-content/uploads/2019/04/Humanity-in-WASH-auditcapacity-Summary-4_3_19-to-CXB.pdf

https://www.pathfinder.org/wp-content/uploads/2022/04/PI-BD_MHM-Rohingya_2022-04-25.pdf

However, no research was done to explore the capacity of local government in delivering WASH services in the host community which was their primary responsibility prior to Rohingya influx. Therefore, we decided to collect data from the host community. In addition, we wanted to avoid duplication of data collection activity as there were multiple agencies who collected regular WASH data from the Rohingya camps. 

Additional comments below and in the attached:

In the findings section there are instances when statistics are provided without reference. “DGHS participants mentioned that 90% of tubewells at the community clinics are not functional.” Specific sources should be provided in all cases where statistics are referenced.

Response: Thank you for this comment. However, this is a primary result from our capacity assessment workshop and mentioned by one of our respondents. Therefore, we cannot use a reference for results. In addition, we cannot verify a statistic provided by the respondent, nor use any reference. 

I recommend that the document is restructured to move the information on the overall institutional arrangements to the introduction- or perhaps develop a background section. Much of the information presented in the “results” section is not specific to Cox’s Bazar and not linked to the data collected through the KIIs or workshops.

Response: Thank you for this comment. However, the overall institutional arrangements is our own interpretation which was generated from discussion with key stakeholders through KII and workshops. This interpretation may not only be specific to Cox’s Bazar and represent the situation of the entire country. Table 2 was generated from our own findings and this is closely linked with the previous paragraphs, so it will not be a good decision to rearrange this section. We hope that you will reconsider your suggestions. 

solid waste is introduced on page 18 for the first time, but it is not included in the table 2. The table should be adjusted or the references to solid waste removed.

Response: Thank you so much for this comment. We have added waste in table 2 after revisiting our raw data.

“Recommendations from the participants” section is poorly organized. It isn’t clear who should be taking forward any specific recommended action, or if there is any general recommendation on changes to the institutional roles and responsibilities that might require a policy or legislative reforms.

Response: Thank you so much for this comment. Recommendations have been separated. 

Reviewer #2: I take this opportunity to thank the authors for preparing this useful report. I have a few suggestions to share

Introduction : Page 5 line 76 the abbreviation DPHE needs to be put in full. In this section it would be useful to highlight more the WASH situation in that region from published literature. I suggest you also highlight how the private sector and other humanitarian organisations are involved WASH in this region seeing you mention them as lacking awareness on WASH policies.

You mention the DPHEs roles, do these involve formulation and awareness creation of the national policies, plans, and frameworks concerning water and sanitation? I think it would be useful background information for the reader to understand.

Response: Thank you so much for this comment. We have elaborated DPHE in the first instance. 

We have included the situation of the Cox’s Bazar on pages 14-16 under “WASH conditions in the context of Cox's Bazar”. We have described a little bit about the current WASH scenario of Cox’s Bazar in the background also. If we want to add more on WASH from published literature, the length of the masnucript will be longer.

Methods

This section is clearly written. A few comments;

The authors reports that a convenient sampling method was used to identify institutions fully or partially involved in WASH-related activities in Cox's Bazar district. I propose a bit more information on how the partial or full involvement was assessed.

Response: Thank you so much for this comment. We have edited the sentence as follows:

“A convenient sampling method was used to identify potential participants from those institutions.”

I suggest the authors correct the referencing errors on page 8 line 152 and 154

Response: Referencing errors corrected. 

Results

In this section is propose the authors set up this into two main sections;

One section to describe the setting/context and another to describe the key themes/ findings from the research activities. As it is its challenging to follow at what point the findings of the interviews begin to be described.

Table 2 a very useful table that describe the key actors. However, its is a rather long table, i suggest some edits to the table to make it easier for the reader to follow.

Response: Thank you so much for this comment. We have edited the results section and Table 2.

I suggest the authors correct the referencing errors on page 8 line 178 and 179

The main results and thematic areas are well described, i am curious to find out if there were any positive findings or reports by the participants from the interviews and workshops. As it stands the authors highlight only the challenges.

Response: Referencing errors corrected. 

Discussion

This section is well written. I propose the authors, set up the discussion to highlight how their findings sit in light of other assessments that have been done in this area or similar settings. Are there things that are unique to this setting ?

I also propose the authors describe /suggest how these WASH policies can be better developed and utilised in such settings seeing there are different actors with different roles.

Response: Thank you so much for this comment. We have edited the discussion section as suggested and also added a conclusion section. 

Include all the abbreviations used in the report in the list of abbreviations

A minor correction on the figures , both are labelled figure 1 and also these figures were not described/highlighted in the main text

Response: List of abbreviations have been suggested by the editor to be removed. 

Figures have been corrected and referenced. 

6. PLOS authors have the option to publish the peer review history of their article (what does this mean?). If published, this will include your full peer review and any attached files.

Do you want your identity to be public for this peer review? For information about this choice, including consent withdrawal, please see our Privacy Policy.

Reviewer #1: No

Reviewer #2: No

---

## [Decision Letter · Decision Letter 1]

3 Nov 2023

PONE-D-23-19851R1Institutional Capacity Assessment in the lens of implementation research: Capacity of the local institutions in delivering WASH services at Cox's Bazar district, BangladeshPLOS ONE

Dear Dr. Rahman,

Thank you for submitting your manuscript to PLOS ONE. After careful consideration, we feel that it has merit but does not fully meet PLOS ONE’s publication criteria as it currently stands. Therefore, we invite you to submit a revised version of the manuscript that addresses the points raised during the review process.

We look forward to receiving your revised manuscript.

Kind regards,

D. Daniel, Ph.D.

Academic Editor

PLOS ONE

**Additional Editor Comments:**

Dear Author,

The 2nd reviewer stated you don't respond to His comments in the pdf file. Please incorporate His comments in your revision. Please let me know if you can't access it.

Reviewers' comments:

Reviewer's Responses to Questions

**Comments to the Author**

1. If the authors have adequately addressed your comments raised in a previous round of review and you feel that this manuscript is now acceptable for publication, you may indicate that here to bypass the “Comments to the Author” section, enter your conflict of interest statement in the “Confidential to Editor” section, and submit your "Accept" recommendation.

Reviewer #1: (No Response)

Reviewer #2: All comments have been addressed

2. Is the manuscript technically sound, and do the data support the conclusions?

Reviewer #1: No

Reviewer #2: Yes

3. Has the statistical analysis been performed appropriately and rigorously? 

Reviewer #1: N/A

Reviewer #2: Yes

4. Have the authors made all data underlying the findings in their manuscript fully available?

Reviewer #1: No

Reviewer #2: Yes

5. Is the manuscript presented in an intelligible fashion and written in standard English?

Reviewer #1: No

Reviewer #2: Yes

6. Review Comments to the Author

Reviewer #1: The authors did not address ANY of the additional comments made in the document in sticky notes. These comments must be addressed.

Most importantly the authors have not sufficiently addressed the most critical comments of my review. Most notably 1) why Cox's was selected (vs any other districts in BGD) and 2) why only focusing on host population verses the entire population is rigorous/insightful.

In their response to these comments the authors simply state that there is already research on the FDMNs and WASH service delivery to that population, HOWEVER they did not adjust the manuscript to reflect the key findings of the existing literature or address the other aspect of my comment (I.e. question #1). I still don’t understand why Cox’s Bazaar was chosen and what (if any) results are more broadly applicable to other districts in Bangladesh? In the abstract the first sentence is “The influx of Forcibly Displaced Myanmar Nationals (FDMNs) has left the Southwest coastal district of Cox's Bazar with one of the greatest contemporary humanitarian crises, stressing the existing water, sanitation, and hygiene (WASH) resources and services.” This suggests to me that indeed, their study cannot ignore the FDMN and needs to incorporate the key findings from existing literature on WASH services to that population into the discussion and recommendations of this work.

In addition, and importantly they reference (in the introduction) a “similar assessment” done through the WASHCost project of institutional mapping. This work also looked a primary data collected on WASH services for the population in question. Why did the authors not choose to include this aspect in their study?

In the manuscript there are statistics referenced with no sources attributed to them (e.g., 90% non-functionality of tubewells). If this is a quotation from a specific participant than it it needs to be identified as such. If there is no additional information to substantiate this statistic (I.e. to independently verify it) than it needs to be explicitly stated as such. Otherwise, there is a risk that your report becomes a reference which could be cited by others to further justify this statistic.

Reviewer #2: I want to thank the authors for making the corrections as suggested. I note the paper now reads much better.

7. PLOS authors have the option to publish the peer review history of their article (what does this mean?). If published, this will include your full peer review and any attached files.

Reviewer #1: No

Reviewer #2: No

---

## [Author Response · Author response to Decision Letter 1]

19 Nov 2023

November 08, 2023

To: Editor, PLOS ONE

Re: Revision and responses to reviewer’s comments

Dear Editor,

Thank you for reviewing our manuscript entitled “Institutional Capacity Assessment in the lens of implementation research: Capacity of the local institutions in delivering WASH services at Cox's Bazar district, Bangladesh” for consideration for publication in the PLOS ONE.

We have made and highlighted revisions by using track changes mode to the manuscript based on editor’s and reviewer’s comments. Here are our point-by-point responses to the comments for kind consideration.

Additional Editor Comments:

Dear Author,

The 2nd reviewer stated you don't respond to His comments in the pdf file. Please incorporate His comments in your revision. Please let me know if you can't access it.

Response: We have edited our previous version of manuscript based on suggestions from both the Reviewers. However, we have not responded to all quarries by Reviewer 1, which we have added in this response document, as well as did additional edits to the manuscript.

Reviewers' comments:

Reviewer's Responses to Questions

Comments to the Author

1. If the authors have adequately addressed your comments raised in a previous round of review and you feel that this manuscript is now acceptable for publication, you may indicate that here to bypass the “Comments to the Author” section, enter your conflict of interest statement in the “Confidential to Editor” section, and submit your "Accept" recommendation.

Reviewer #1: (No Response)

Reviewer #2: All comments have been addressed

2. Is the manuscript technically sound, and do the data support the conclusions?

Reviewer #1: No

Reviewer #2: Yes

3. Has the statistical analysis been performed appropriately and rigorously? 

Reviewer #1: N/A

Reviewer #2: Yes

4. Have the authors made all data underlying the findings in their manuscript fully available?

Reviewer #1: No

Reviewer #2: Yes

5. Is the manuscript presented in an intelligible fashion and written in standard English?

Reviewer #1: No

Reviewer #2: Yes

6. Review Comments to the Author

Reviewer #1: 

The authors did not address ANY of the additional comments made in the document in sticky notes. These comments must be addressed.

Response: We apologies for this error; we have addressed all comments in the previous revision, but we have not provided responses. We added all responses to this round. However, edits were submitted in previous rounds and also kept in this revision.

Most importantly the authors have not sufficiently addressed the most critical comments of my review. Most notably 1) why Cox's was selected (vs any other districts in BGD) and 2) why only focusing on host population verses the entire population is rigorous/insightful.

In their response to these comments the authors simply state that there is already research on the FDMNs and WASH service delivery to that population, HOWEVER they did not adjust the manuscript to reflect the key findings of the existing literature or address the other aspect of my comment (I.e. question #1). I still don’t understand why Cox’s Bazaar was chosen and what (if any) results are more broadly applicable to other districts in Bangladesh? In the abstract the first sentence is “The influx of Forcibly Displaced Myanmar Nationals (FDMNs) has left the Southwest coastal district of Cox's Bazar with one of the greatest contemporary humanitarian crises, stressing the existing water, sanitation, and hygiene (WASH) resources and services.” This suggests to me that indeed, their study cannot ignore the FDMN and needs to incorporate the key findings from existing literature on WASH services to that population into the discussion and recommendations of this work.

In addition, and importantly they reference (in the introduction) a “similar assessment” done through the WASHCost project of institutional mapping. This work also looked a primary data collected on WASH services for the population in question. Why did the authors not choose to include this aspect in their study?

Response: Thank you for your comment. We have critically analyzed this and we realized that mentioning “Host community” was not appropriate here. Those institutions are responsible for entire Cox’s Bazar district including FDMN and host population; therefore, geographic area for this paper is entire Cox’s Bazar district. We have deleted “Host Community” from the entire manuscript.

We have already discussed in the introduction that why Cox’s Bazar is relevant. As the Rohingya influx increased the burden of local DPHE in Cox’s Bazar, so we conducted an assessment there. The burden of DPHE in other districts is same, and have not increased much compared to Cox’s Bazar. This has been discussed in the introduction section, last sentence of first paragraph and the entire second paragraph. Also, part of third paragraph is explain this.

In addition to that, we have collected primary data from the Cox’s Bazar district, but we have not added those data here as this will significantly increase the size of the manuscript and reader will lost interest to read this long paper. We have prepared another manuscript from the primary data collected from households, schools, hospitals, and public places. 

In the manuscript there are statistics referenced with no sources attributed to them (e.g., 90% non-functionality of tubewells). If this is a quotation from a specific participant than it it needs to be identified as such. If there is no additional information to substantiate this statistic (I.e. to independently verify it) than it needs to be explicitly stated as such. Otherwise, there is a risk that your report becomes a reference which could be cited by others to further justify this statistic.

Response: Thank you for this comment. This data came from the respondent, so there is no source to cite. We have edited the sentence as below:

“DGFP own study findings showed that nearly 40-50% of facilities are unhygienic. DGHS participants mentioned that approximately 90% of tubewells at the community clinics are not functional.”

Reviewer #2: I want to thank the authors for making the corrections as suggested. I note the paper now reads much better.

7. PLOS authors have the option to publish the peer review history of their article (what does this mean?). If published, this will include your full peer review and any attached files.

Do you want your identity to be public for this peer review? For information about this choice, including consent withdrawal, please see our Privacy Policy.

Reviewer #1: No

Reviewer #2: No

 

Comments from Reviewer 1 in previous round

1. Underperforming historically? please clarify the timeframe here.

Response: The overall WASH situation in Cox’s Bazar was below average compare to other districts from the independence. Therefore, we have added “historically”.

2. Should specify upazilas.

Response: Thank you, edited as “especially in the Ukhiya and Teknaf Upazila”

3. This health bulletin from WHO covering a few weeks hardly seems to be an appropriate resources to support this statement.

Response: Thank you; other study citation added.

4. Why only host community? FDMN population in the relevant upazilas is much larger. After reading the first few pages of this, I still didn't get what the geographic scope of the work is and if it includes (or excludes) FDMN and the WASH systems, services, and relevant stakeholders. 

Response: Thank you for your comment. We have critically analyzed this and we realized that mentioning “Host community” was not appropriate here. Those institutions are responsible for entire Cox’s Bazar district including FDMN and host population; therefore, geographic area for this paper is entire Cox’s Bazar district. We have deleted “Host Community” from the entire manuscript. 

5. And to all people

Response: added “and to all people”

6. Remove "although"

Response: Removed.

7. This WASH cost paper is presenting work which is much broader than an institutional capacity assessment, but a much more broad "institutional mapping" which included case studies and primary data collected through a rapid assessment. 

Response: That is why we cited this to establish the point of robust institutional assessment instead of rapid assessment

8. What is the reason that there weren't key informant interviews from a representative of each group?

Response: We have not found anyone for KII from those group during our data collection in particular Upazila.

9. This was a KII with an NGO representative?

Response: Private sector representative

10. This should be further disaggregated- at least into NGO/civil society and private sector

Response: Thank you, segregated.

11. With the exception of the 7th group which mixed civil society and private sector. If the assumption was that it is necessary to separate the groups, then you need to explicitly address why you chose not to separate the groups in the 7th workshop. and also what the limitations might be of doing so.

Response: There were not enough participants to conduct a separate workshop with private sector representatives, dealers, and entrepreneurs. The number of water and sanitation products manufacturer are not too many in Cox’s Bazar. 

12. These are not the results of the study. this is general information on Bangladesh and should be moved to the introduction.

Response: Omitted.

13. This paragraph is also stating information which is not specific to Cox's Bazar and the primary data collected therein. Therefore, unless there is information that is unique to the institutional arrangements in Cox's Bazar, I would suggest it is moved out of the results section and into the introduction/background

Response: Local institutions and programs related to WASH service delivery in Cox's Bazar and the structure of the WATSAN Committee narrative developed from our primary data; so, we would prefer to keep this here to understand the next sections of the results. Though we agree with the reviewer that this information may be similar for other districts.

14. Are these responsibilities different for Cox's Bazar as opposed to other areas? in what way? why? if you are unable to answer these questions, than I would ask- why are we focuing on Cox's Bazar specfiically, verses any where else in Bangladesh? One answer coudl be because of the influx of displaced persons. However, that is not addressed/discussed in the institutional mapping....

Response: This responsibility is similar for other Upazila in Bangladesh. However, as the Rohingya influx increased the burden of local DPHE in Cox’s Bazar, so we conducted an assessment there. This has been discussed in the introduction section, last sentence of first paragraph and the entire second paragraph. Also, part of third paragraph is explain this.

15. Seems this table could be improved, edited to be more concise but also to convey more information. see example from WASH cost paper you cited. Box 1 on page 7- https://cess.ac.in/wp-content/uploads/2019/12/CESS-WASHCost-Working-Paper-No.04.pdf

Response: Thank you so much for your suggestions. We developed this table for Cox’s Bazar only, so we have not highlighted central level responsibilities. Also, the central level institutional responsibilities are different, as also noticed from the WASH cost paper. Therefore, we keep the table in same format with some revision according to your other suggestions. We hope that, this is acceptable.

16. Is there a need to separate out MOLGRD&C? to understand their role. Not sure if DPHE fulfills all functions related to WASH that MOLGFD&C is involved in...

Response: DPHE (Department of Public Health Engineering) is the implementing department under the Ministry of Local Government, Rural Development & Co-operatives (MOLGRD&C). As there is no specific role of MOLGRD&C as a separate agency in Cox’s Bazar, and this is a central level agency, so we have not separated it.

17. Who is in charge of on-going testing?

Response: DPHE Cox’s Bazar.

18. Similar comment to the above

Response: Same as above response; Directorate of Primary Education is under the Ministry of Primary and Mass Education and implement in Cox’s Bazar. 

19. No responsibility for water supply in HCF?

Response: Thank you for this suggestion, added

20. No responsibility for sanitation services in HCF?

Response: Thank you for this suggestion, added

21. This is not a responsibility

Response: Thank you for this suggestion, Omitted.

22. Does this need to be divided into two rows, one for Upazila and one for Union?

Response: Thank you for this suggestion, we have added WATSAN as separate committee which work in union level.

23. NGOs are included here (which is the standout amongst the government institutions in the list). It seems like if NGOs are listed then there should be other service provider groups listed- private sector, school management committee, etc

Response: Thank you for this suggestion, edited.

24. it seems like there is superfluous information included in this section. as opposed to listing examples the table should be organized by sub-sector (water, sanitation, hygiene) and explicitly state the role of NGOs in those subsectors. Some examples can be provided, but should be concise and where clarity is needed.

Response: Thank you for this suggestion, edited.

25. WATSAN committees?

Response: Thank you for this suggestion, added WATSAN committee.

26. What study?

Response: The respondent referred to DGFP own study.

27. This is very precise, where is this data coming from? please cite source (e.g. is this a DGHS database that they maintain, a specific report, etc)

Response: This data came from the respondent, so there is no source to cite. We have edited the sentence as below:

“DGFP own study findings showed that nearly 40-50% of facilities are unhygienic. DGHS participants mentioned that approximately 90% of tubewells at the community clinics are not functional.”

28. Why? what is main cause of non-functionality?

Response: We have described this in another paper which is forthcoming, in this paper we have only described institutional settings, responsibilities, and their viewpoints. From our analysis, we can say that, there is no or very low maintenance, so most of the tubewells are non-functional. But we have not added this results in this paper.

29. Cox's bazar is a on a peninsula with a lot of water supply issues. is this specific to one area? do declining water levels represent a majority of the problems linked to non-functionality?

Response: Thank you for this comment. After the Rohingya influx, nowadays this is a major problem in some areas, especially hilly area. This is not specific to any particular Upazila. Non-functionality is not linked with declining water levels.

30. Does this damage them? or make users less likely to use them?

Response: High concentration of iron in the water also leaves a stain in the sanitary ware, and damaging them.

31. Don't capitalize

Response: Thank you, edited

32. Consider the implications of the wording. DSHE reported or DSHE attributed....verses saying that lack of facilities caused the absence. It is also possible that the absence was caused by lack of materials/supplies or social stigma or all of the above. Unless there is a conclusive study by DSHE, and the participants are providing their (professional) opinions, it should be phrased so that the reader understands as such. 

Response: Thank you for this suggestion. We have edited the text as below:

“Lack of menstrual hygiene management (MHM) facilities at school toilets may have been a reason behind the absence of female students during menstruation, as attributed by DSHE participants.”

33. Please clarify what this means "inadequate manufacturing of need. based sanitary ware or products that are appropriate for different geographical contexts" Is this talking about latrine designs? 

Response: This talks about latrine designs that are manufactured differently with different needs in consideration, for example, a higher latrine for flood prone areas. We have edited the text as below:

“Private sector participants mentioned that there is inadequate availability of need-based sanitary ware or products that are appropriate for different geographical contexts.”

34. Durable? if you say "sustainable" you will need to define it.

Response: Thanks, edited.

35. Choose a different word than "distribute"

Response: Thanks, edited.

36. what does this mean? compromised in what way?

Response: Thanks, omitted.

37. Construction? drilling?

Response: Thanks, edited as construction.

38. I don't quite follow. What is done with the respective sums of money received by each? please clarify what is meant in this quote.

Response: The allocation itself is for the distribution of tubewells. 50% of that allocation goes to the Upazila Chairman for tubewell installation through WATSAN committee, and the rest 50% goes to MP.

39. It seems like the previous paragraphs were suggesting a "top down approach". if not, then it needs to be clarified.

Response: Edited as “Also, this type of top-down approach suggested for the budgeting system is not always context-specific.”

40. "Also" suggests that the previous paragraph does NOT have a conclusion. Is there an absence of effective coordination? or not? the KI suggested that there is regular connection (and implicitly coordination) regardless of the lack of formal meetings. this needs to be clarified

Response: Thank you for this comment. This was initially written "also" since the beginning of the last paragraph also started by mentioning a lack of coordination. Now we have edited as:

“We found inadequate coordination between the Upazila and Union WATSAN Committees or Union Standing Committee.”

41. solid waste is introduced here for the first time. I suggest that you revisit the table above since solid waste managment is not included in the institutional responsibilities.

Response: Thank you for this comment. We mentioned solid waste in the table 2 under DGHS responsibilities:

“Monitor waste management at marketplaces including slaughterhouse and fish markets. In addition, they monitor food safety and hygiene of food courts and food shops.”

42. "Wash specific" or do you mean WASH in specific contexts?

Response: We mean wash specific; this means activities that target WASH outcomes.

43. "secondary eduction" is not an activity

Response: Thank you for this comment. We have modified:

“Participants also reported not having a specific budget allocation to implement WASH-specific activities (e.g., for providing inclusive wash blocks in secondary education institutes).

44. What does "counseling activity" mean? please clarify

Response: Counseling activities related to issues around WASH service and practices at the community level. We have modified:

“Frontline workers/community health workers do activities related to issues around WASH service and practices at the community level (e. g. courtyard session, one-to-one session etc).”

45. Who are these recommendations for? for example, DPHE recommends including disabled in planning. but WHO should be doing that? it seems more logical that the recommendations (if they are on specific action to be taken) should be divided into those for specific institutions. 

you could also separate these into general recommendations on institutional reform. 

as it stands it is a laundry list of things that people said. 

Response: All the recommendations seem to come from the DPHE and some from the private sector. Hence, we have separating them by DPHE and Private sector. 

46. More of less?

Response: Similar.

47. How does this improve the capacities?

Response: We have identified the gaps here, so if all the organizations fill those gap out- then the capacities will improve.

48. There was very little private sector engagement in this research

Response: Currently, there is no involvement of private sector in WASH Sector Planning. We have identified this gap also, so we are recommending to engage more.

49. But this information was not presented, why?

Response: Thank you for asking. This paper is only representing the institutional capacity, and we have not presented any data from the household level. If we add household level data, this will be long paper, and people will not read the entire paper. We are developing another manuscript for availability of WASH infrastructure in Cox’s Bazar.

---

## [Decision Letter · Decision Letter 2]

27 Dec 2023

Institutional Capacity Assessment in the lens of implementation research: Capacity of the local institutions in delivering WASH services at Cox's Bazar district, Bangladesh

PONE-D-23-19851R2

Dear Dr. Rahman,

We’re pleased to inform you that your manuscript has been judged scientifically suitable for publication and will be formally accepted for publication once it meets all outstanding technical requirements.

Kind regards,

D. Daniel, Ph.D.

Academic Editor

PLOS ONE

Additional Editor Comments (optional):

Reviewers' comments:

Reviewer's Responses to Questions

**Comments to the Author**

1. If the authors have adequately addressed your comments raised in a previous round of review and you feel that this manuscript is now acceptable for publication, you may indicate that here to bypass the “Comments to the Author” section, enter your conflict of interest statement in the “Confidential to Editor” section, and submit your "Accept" recommendation.

Reviewer #1: All comments have been addressed

2. Is the manuscript technically sound, and do the data support the conclusions?

Reviewer #1: Yes

3. Has the statistical analysis been performed appropriately and rigorously? 

Reviewer #1: N/A

4. Have the authors made all data underlying the findings in their manuscript fully available?

Reviewer #1: No

5. Is the manuscript presented in an intelligible fashion and written in standard English?

Reviewer #1: Yes

6. Review Comments to the Author

Reviewer #1: The authors addressed all comments. The only remaining issues is in regard to the data transparency and compliance with the PLOS data policy. I will leave this issue to the editor.

7. PLOS authors have the option to publish the peer review history of their article (what does this mean?). If published, this will include your full peer review and any attached files.

Reviewer #1: **Yes: **Ryan W Schweitzer

---

## [Editor Report · Acceptance letter]

25 Jan 2024

PONE-D-23-19851R2 

PLOS ONE

Dear Dr. Rahman, 

I'm pleased to inform you that your manuscript has been deemed suitable for publication in PLOS ONE. Congratulations! Your manuscript is now being handed over to our production team.

Kind regards, 

on behalf of

Mr D. Daniel 

Academic Editor

PLOS ONE